# One-Shot Style Personalization for RL Agents via Latent Discriminator

## Abstract

Reinforcement learning (RL) has achieved remarkable success in training agents with high-performing policies, and recent works have begun to address the critical challenge of aligning such policies with human preferences. While these efforts have shown promise, most approaches rely on large-scale data and do not generalize well to novel forms of preferences. In this work, we formalize one-shot style alignment as an extension of the preference alignment paradigm. The goal is to enable RL agents to adapt to human-specified styles from a single example, thereby eliminating the reliance on large-scale datasets and the need for retraining. To achieve this, we propose a framework that infers an interpretable latent style vector through a learned discriminator and adapts a pretrained base policy using a style reward signal during online interaction. Our design enables controllable and data-efficient alignment with target styles while maintaining strong task performance, and further enables smooth interpolation across unseen style compositions. Experiments across diverse environments with varying style preferences demonstrate precise style alignment, strong generalization, and task competence.

## 1 Introduction

Reinforcement learning (RL) has achieved remarkable success in continuous control and multi-agent domains Chen et al. (2023); Schrittwieser et al. (2020). Despite these advances, aligning agent behavior with nuanced, human-preferred styles remains a fundamental challenge. Focusing exclusively on task reward optimization often leads agents to converge toward undifferentiated policies, thereby neglecting stylistic variability Gabriel (2020). Human preferences are inherently multi-dimensional and involve trade-offs (e.g., speed versus safety), rendering the alignment problem analogous to multi-objective optimization Hemphill (2020). Traditional RL approaches, focused on single-objective reward maximization, struggle to capture this complexity, limiting their applicability in human-facing settings.

Consider autonomous taxi driving as a concrete example. An online-trained agent that optimizes a reward function emphasizing efficiency may adopt a "balanced" driving style. In reality, passengers exhibit diverse preferences: some prioritize cautious driving for comfort and safety, while others prefer assertive driving to minimize travel time. A single, fixed policy cannot satisfy this diversity, highlighting the necessity for personalized fine-tuning mechanisms. Figure 1 illustrates four representative driving styles—*Safety*, *Efficiency*, *Comfort*, and *Economy*—demonstrating that preferences vary not only in style but also in intensity. Capturing such continuous and nuanced variations requires methods capable of fine-grained, personalized alignment. Beyond driving, similar challenges manifest in domains such as medical decision-making Reverberi et al. (2022); Yu et al. (2024) and home service robotics Gonzalez-Aguirre et al. (2021), where alignment with human values is indispensable.

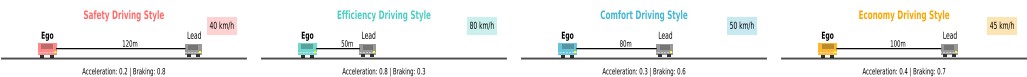

Figure 1: Illustration of different autonomous driving styles. Each style exhibits distinct speed, acceleration, braking, and following distance, highlighting the need for personalized alignment.

However, most existing alignment frameworks, such as those based on supervised learning, imitation learning, or preference modeling Gabriel (2020); Noothigattu et al. (2018), treat preferences as singular and static objectives. This one-dimensional formulation neglects both the diversity and the degree of individual preferences, limiting personalization and raising fairness and safety concerns in sensitive applications Yu et al. (2024). Under these constraints, adapting an agent to a different preference typically requires substantial additional data and retraining, making the process costly and inflexible. Furthermore, existing approaches face significant challenges in effectively incorporating style preferences into RL policies: reward-based methods either provide sparse trajectory-level signals that are too weak to guide nuanced behaviors or apply dense reward shaping that can conflict with primary task objectives Christiano et al. (2017); imitation learning requires large quantities of style-specific demonstrations, which are expensive to collect and annotate Pandy et al. (2025); offline RL can leverage pre-collected trajectories, but distributional shift and conservative value estimation limit its effectiveness. In addition, most methods model style as discrete labels or fixed objectives, overlooking its continuous and multi-dimensional nature. These limitations collectively motivate the development of a framework that can flexibly and efficiently encode style preferences, capture their nuanced structure, and support adaptation without extensive retraining or additional data.

To overcome these limitations, we propose a two-stage framework for personalized behavioral alignment. Firstly, we train a task-competent base policy using standard RL algorithms (e.g., SAC for single-agent control, PPO for multi-agent cooperation). Next, we construct an interpretable latent style space using a discriminator based on granular-ballXia et al. (2019) trained on a multi-style trajectory dataset. Unlike conventional classifiers, this discriminator partitions trajectories into robust and semantically meaningful clusters, yielding style vectors that capture axes such as "speed vs. caution" and "efficiency vs. liveliness." These continuous vectors not only identify distinct styles but also quantify the degree of preference along each semantic axis, enabling fine-grained personalization. The granular-ball mechanism enhances interpretability and robustness, ensuring stable style encoding even under noisy or ambiguous trajectories.

With the latent style space established, policy adaptation to new preferences becomes both straightforward and efficient. A single style-specific demonstration is sufficient to infer a target style vector, which then conditions the policy as input to the reward model, enabling one-shot alignment without full retraining. By exploiting the compositional structure of the style space, the policy can generalize to previously unseen style combinations without additional training, enabling zero-shot adaptation and reducing reliance on extra data.

We evaluate our framework on both single-agent MuJoCo locomotion (Hopper) and multi-agent cooperative tasks (Overcooked). Results show that our approach enables one-shot adaptation, accurately reproduces target styles, and outperforms baselines in alignment accuracy, style fidelity, and generalization. Our main contributions are:

- We design a low-dimensional, interpretable latent style space learned via a granular-ball discriminator, supporting compositionality and interpolation of behavioral styles.

- We introduce a fast adaptation mechanism using a single style demonstration, enabling one-shot style alignment without full retraining.

- We conduct extensive experiments across single- and multi-agent tasks, demonstrating superior performance in style fidelity, task reward, and alignment efficiency compared to baselines.

## 2 RELATED WORKS

### 2.1 ALIGNING AI WITH HUMAN VALUES

A long-standing challenge in intelligent agent design is ensuring that agents act in accordance with human preferences and ethical norms. Early research primarily focused on universal value alignment. Ng et al. Ng et al. (2000) introduced inverse reinforcement learning (IRL) to infer latent human values from demonstrations, laying the foundation for cultural adaptation. However, such approaches often rely on large-scale, high-quality datasets Adams et al. (2022). Subsequent work expanded the paradigm: Hadfield-Menell et al. Hadfield-Menell et al. (2016) formalized cooperative

imitation, while Li et al. Li et al. (2015) proposed mentor-based learning to capture procedural alignment cues. More recently, Christiano et al. Christiano et al. (2017) advanced reward modeling from human feedback, enabling alignment in high-dimensional domains. Reinforcement Learning from Human Feedback (RLHF) has since become central to fine-tuning large language models Ouyang et al. (2022). Beyond language, preference-based alignment has also been explored in domains such as games Dong et al. (2023), and Liu et al. Liu et al.(2025) demonstrated alignment of pre-trained agents to human styles by calibrating a trajectory-level reward model and gradually transitioning from environment to preference rewards via a linear curriculum.

As alignment objectives shift from population-wide ethics to individual preferences, the challenge of personalization has become increasingly prominent. Fan et al. Fan & Poole (2006) highlighted the difficulty of dynamically adapting to users under system constraints, while Pearce et al. Pearce et al. (2023) noted the challenge of capturing diverse and evolving individual preferences. One promising direction involves mapping natural language instructions into reward functions Xue et al. (2023); Swamy et al. (2023); Akrour et al. (2011); Hadfield-Menell et al. (2016), although this often depends on precise and expressive communication. To address this limitation, Guan et al. Guan et al. (2022) decomposed behaviors into relative attributes, thereby supporting more intuitive preference specification.

Imitation learning provides another pathway to alignment. By leveraging demonstrations Hussein et al. (2017), agents can adapt to human-like behaviors without explicit reward design. Torabi et al. Torabi et al. (2018) further extended this idea by proposing a self-supervised observation-only framework, eliminating the need for action labels and accelerating adaptation.

Despite these advances, existing approaches remain heavily dependent on large datasets and struggle to generalize across styles or domains. They typically require training a separate agent for each target style, which incurs prohibitive time and computational costs.

## 2.2 Granular Ball Methods for Structured Representation

Granular Ball (GB) methods stem from Granular Computing (GrC) theory, emphasizing multi-granularity and interpretable data abstractionXia et al. (2019). GB represents local structures via adaptive hyperspheres ("balls") built under purity and compactness constraints Zhang et al. (2023); Cheng et al. (2024); Sun et al. (2025). Unlike traditional clustering (e.g., K-Means), GB prioritizes local consistency and overlap minimization, making it well-suited for handling complex, uneven distributions.

Recent work has applied GB to tasks like feature selection and concept learning, demonstrating strong robustness. In RL and imitation learning, GB structures offer a principled way to cluster trajectories by stylistic consistency, enabling interpretable latent spaces and controllable behavior modeling—especially when extended with kernelization to capture nonlinear separability in high-dimensional trajectory spacesLiu et al. (2024).

## 3 Method

Let $\mathcal{M} = (\mathcal{S}, \mathcal{A}, P, r, \gamma)$ be a Markov Decision Process (MDP) representing the agent's task (e.g., locomotion or cooperative delivery), where $\mathcal{S}$ is the state space, $\mathcal{A}$ is the action space, $P(s'|s, a)$ denotes the probability of transitioning to state $s'$ after taking action $a$ in state $s$, $r(s, a)$ is the immediate reward received for this transition, and $\gamma \in [0, 1]$ is a discount factor balancing immediate and future rewards. We assume a pretrained base policy $\pi_0(a|s)$ that is competent on the task but lacks any particular style. Our goal is to adapt $\pi_0$ so that its behavior matches a target style provided by a single demonstration trajectory $\tau^* = (s_t, a_t, r_t)_{t=0}^{T}$ collected by a human or expert agent.

The overall pipeline is summarized as follows: (1) we infer a latent style vector $z_{\text{target}}$ from the demonstration using a discriminator built on Granular Ball (GB) clustering; (2) during fine-tuning, we compute the style reward in real time by embedding the agent's current trajectory segments and measuring their similarity to $z_{\text{target}}$, ensuring that the reward signal reflects the evolving behavior of the policy; (3) we dynamically balance environment and style rewards to preserve task performance while promoting alignment; and (4) we enable interpolation in the latent space to generalize beyond

predefined styles. The framework and process of our method are illustrated in Figure 2. A full pseudocode of the alignment procedure is provided in Appendix G.3 for clarity and reproducibility.

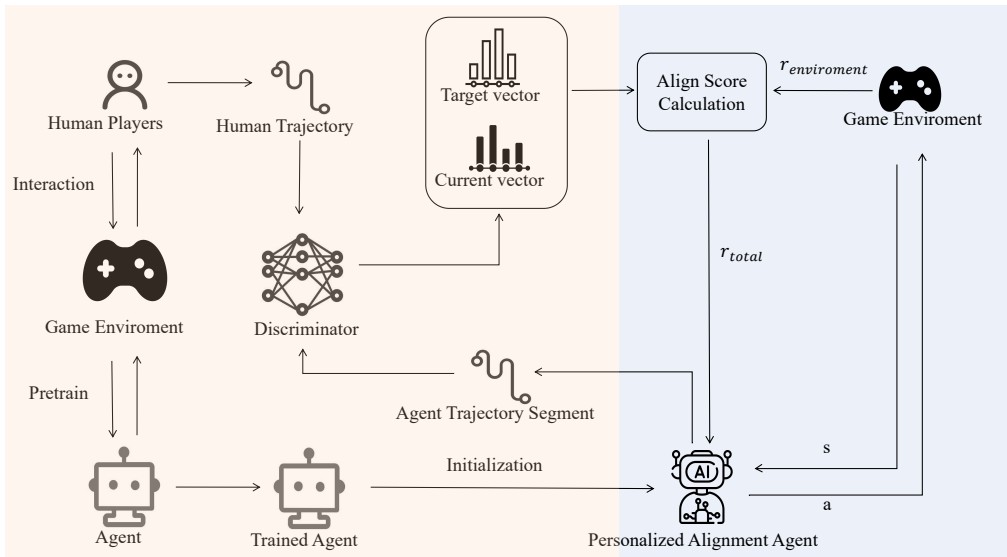

Figure 2: Overview of the reward model. The left (orange) section illustrates single-sample style reward computation, including style similarity evaluation. The right (blue) section depicts the style alignment process, showing how individual style rewards are integrated to guide policy adaptation.

### 3.1 LATENT STYLE REPRESENTATION VIA DISCRIMINATIVE EMBEDDING

We model human-preferred behavioral styles by embedding trajectories into a continuous latent space using a style discriminator, which produces pseudo-labeled style predictions for reward shaping and policy conditioning. In the absence of annotated style labels, trajectories are grouped into style-consistent clusters via the Granular Ball (GB) method, with each cluster serving as a pseudo-style category. Trajectories are represented by descriptors capturing temporal, kinematic, and spectral characteristics, all standardized for stable training. We adopt GB clustering instead of traditional K-means due to its flexibility and robustness. GB can adaptively partition data at multiple granularities without predefining the number of clusters, handle complex and non-convex distributions, and resist noise or outliers. Each GB center serves as a style prototype, from which a random forest classifier generates a soft assignment vector, and the latent style vector is constructed from inverse distances to GB centers, normalized on the probability simplex.

The resulting latent space is continuous rather than restricted to discrete clusters, enabling fine-grained control over behavior. A target style vector $\mathbf{z}_{\text{target}} \in \mathbb{R}^d$ can take any normalized value ($\|\mathbf{z}_{\text{target}}\|_2 = 1$), allowing smooth interpolation between base styles as well as extrapolation to unseen mixtures. This flexibility supports style-aware reward shaping and makes the framework scalable to human demonstrations: although our experiments use pseudo-style labels derived from handcrafted agent behaviors, the same mechanism can directly induce representative styles from human data, synthesizing hybrid or sharpened styles tailored to specific preferences. Further implementation details, including the pseudocode for the GB-based discriminator, are provided in Appendix G.1.

### 3.2 REWARD MODEL CONSTRUCTION

The style reward is computed by quantifying the similarity between the agent's current style and the target style. We use cosine similarity to measure the closeness between the style representation of the current trajectory and that of the target style. The style reward is then calculated as follows:

$$R_{\text{style}} = \alpha \cdot (\text{COS}(x, y))^2 \tag{1}$$

where $\alpha$ is a scaling factor to adjust the influence of the style similarity, and $\mathrm{COS}(x, y)$ represents the cosine similarity between the feature representations of the current and target trajectories. Squaring the similarity emphasizes higher similarity scores while diminishing lower ones, promoting stronger alignment with the target style.

To prevent the style reward from dominating the learning process and undermining task performance, we introduce a **dynamic weighting mechanism** to balance environment rewards and style rewards. Specifically, the weight assigned to the style reward, $s_w$, gradually increases during training. The weight is defined as:

$$s_w = w_{\min} + (w_{\max} - w_{\min}) \cdot \frac{c - c_{\min}}{c_{\max} - c_{\min}} \tag{2}$$

Here:

- $c$: the current training step.
- $c_{\min}$ and $c_{\max}$: the minimum and maximum training steps.
- $w_{\min}$ and $w_{\max}$: the minimum and maximum weights for the style reward.

This approach allows the model to focus more on completing tasks effectively during early training stages and gradually increase its emphasis on style alignment as training progresses.

Finally, the overall reward signal is computed by combining the environment reward with the dynamically weighted style reward:

$$R_{\text{total}} = R_{\text{environment}} + s_w \cdot R_{\text{style}} \tag{3}$$

where $R_{\text{total}}$ is the final reward signal, $R_{\text{environment}}$ is the reward directly derived from the environment, and $s_w$ is the dynamic weight for the style reward. This formulation ensures a balanced and progressive alignment between task execution and style adherence.

## 4 EXPERIMENTS

We conduct experiments in two representative environments—Hopper (MuJoCo) and Overcooked—to evaluate the effectiveness of our style alignment framework. Our evaluation focuses on (i) the accuracy of the latent style discriminator, and (ii) the alignment performance of the personalized agent across predefined stylistic behaviors.

### 4.1 EXPERIMENTAL SETUP

**Environments.** To examine the generalization ability of our latent style discriminator (LSD) across different control granularities, we conduct experiments in two representative environments. For low-level control behaviors, we utilize `Hopper-v5` from the MuJoCo suite, a single-legged robot with three controllable joints (hip, knee, ankle) that performs locomotion in continuous action and high-dimensional state spaces. For high-level strategic coordination, we use the `Overcooked-AI` environment, a multi-agent gridworld domain designed for human-AI collaboration in partially observable cooking tasks.

**Style Definition.** We abstract real-world behavioral complexity into a finite set of high-discriminability styles for each environment. Four styles are chosen per environment to balance representativeness and discriminability: they are sufficiently distinct to capture diverse behavioral patterns, while remaining manageable for controlled experiments and clear evaluation. In Hopper, four representative locomotion styles are defined via targeted reward shaping: Smooth-Motion, Energy-Saving, Posture-Stable, and Speed-Oriented. In Overcooked, policies are categorized into four strategic styles: Selfish, Cooperative, Enjoyable, and Efficient. These styles capture meaningful, human-interpretable patterns across both low-level motor control and high-level strategic interaction. For implementation details and visual examples of each style, we refer readers to Appendix C.

**Implementation Details.** Experiments were conducted on a multi-GPU Linux server with multiple random seeds to ensure reproducibility. Additional information on hardware, training setup, style reward weighting, and data collection procedures is provided in Appendices E and B.

**Evaluation Metrics.** We adopt a two-stage evaluation protocol to assess both the quality of style modeling and the effectiveness of downstream policy alignment. For the **style discriminator**, we report classification loss, accuracy, and macro-averaged F1 score on the train/validation/test splits, ensuring both overall predictive performance and balanced recognition across style categories. For the **aligned policy**, we employ two complementary metrics: (1) **task reward**, which quantifies the policy's ability to achieve environment objectives, and (2) **style consistency**, computed as the cosine similarity between the predicted and target style vectors. These metrics are chosen to jointly capture functional competence and stylistic adherence—two equally critical aspects in human-aligned control scenarios—thus providing a holistic evaluation of the learned policy.

## 4.2 EVALUATION OF THE LATENT STYLE DISCRIMINATOR

We first assess the performance of the proposed style discriminator on a held-out test set, focusing on the *trajectory segment classifier* that enables early and localized style recognition.

Table 1: Performance of the trajectory segment classifier.

| Trajectory Segment Classifier | | | | |
|---|---|---|---|---|
| Label | Precision | Recall | F1-score | Support |
| 0 | 0.98 | 0.97 | 0.98 | 10,000 |
| 1 | 0.97 | 0.99 | 0.98 | 10,000 |
| 2 | 0.95 | 0.98 | 0.96 | 10,000 |
| 3 | 0.99 | 0.94 | 0.96 | 10,000 |
| Accuracy | | 0.97 | | 40,000 |
| M Avg | 0.97 | 0.97 | 0.97 | 40,000 |
| W Avg | 0.97 | 0.97 | 0.97 | 40,000 |

As shown in Table 1, the discriminator achieves consistently high precision and recall across all four style classes, yielding an overall accuracy of 97%. This demonstrates strong generalization and fine-grained discriminability, validating its effectiveness for multi-style recognition.

Beyond global separability, it is also essential to handle intra-class diversity and stylistically ambiguous trajectories. To this end, we adopt a GB-based discriminator that explicitly models local structure while preserving classification robustness.

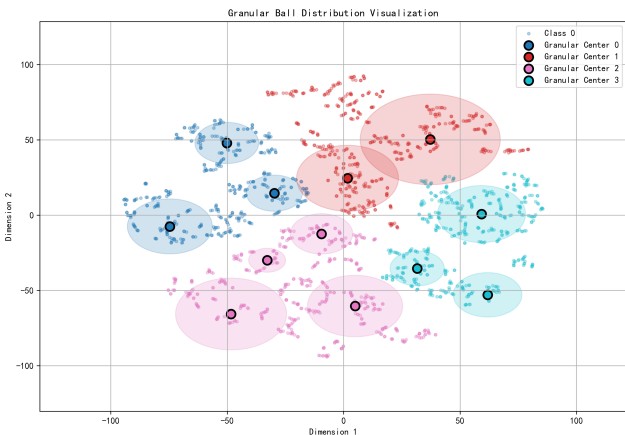

Figure 3: Granular Ball clustering in the latent style space. Each cluster reflects a distinct style with minimal overlap, while transitional zones capture hybrid behaviors.

As illustrated in Figure 3, the resulting embedding space exhibits compact, well-separated clusters along with transitional zones that correspond to hybrid-style behaviors. A hierarchical organization

also emerges, suggesting semantic relationships among styles. This structured latent space supports smooth style transitions and consistent recognition, providing a solid foundation for personalized alignment and style-conditioned policy learning. Further visualization results (t-SNE plots) are provided in Appendix F.1.

## 4.3 PERSONALIZATION AND ALIGNMENT PERFORMANCE

To comprehensively evaluate the effectiveness of our personalized style alignment framework, we conduct comparative experiments in the Hopper environment of MuJoCo. We compare our method (denoted as **Fine-tune RL**) against two baselines: (1) **Behavior Cloning (BC)** and (2) the unmodified **Original RL** policy. The experiment targets four representative styles (Style 0 to Style 3), with average episode rewards under each training regime shown in Figure 4.

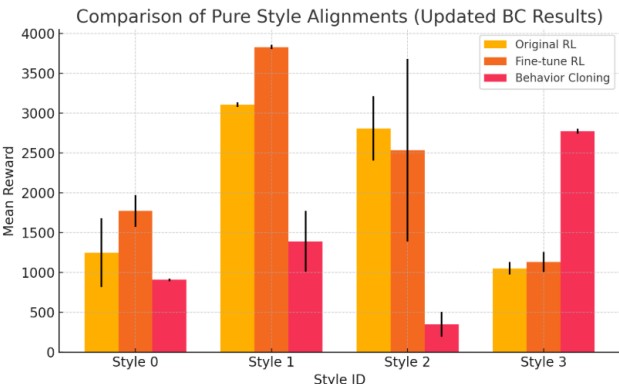

Figure 4: Average episode reward comparison across training methods for each style.

Compared to Behavior Cloning (BC), Fine-tune RL consistently yields higher rewards with lower variance, especially for energy-efficient and posture-stable styles. BC performs competitively on simpler deterministic styles (e.g., speed-focused) but lacks generalization and requires retraining for each style. In contrast, Fine-tune RL enables multi-style adaptation within a single policy through conditioning and reward alignment, offering more robust and scalable control.

## 4.4 STYLE INTERPOLATION AND ALIGNMENT ANALYSIS

To evaluate the controllability and interpretability of the learned latent style space, we perform interpolation between base style vectors. Let $\mathbf{s}_i$ and $\mathbf{s}_j$ denote the one-hot encodings of two base styles. Interpolated vectors are constructed as $\mathbf{s}_{mix} = \mathbf{s}_i + \mathbf{s}_j$, followed by normalization, enabling smooth transitions between styles.

**Single-style Baselines.** The performance of individual styles is reported in Appendix F.2 and serves as reference for mixed-style evaluations.

**Multi-style Combination Results.** We evaluate six binary style combinations (e.g., $[1, 1, 0, 0]$) under two training schemes: multi-task learning and style-aligned finetuning. As shown in Figure 5, finetuning consistently improves task reward and reduces variance, particularly for combinations involving conflicting base styles (e.g., $[0, 0, 1, 1]$), with improvements exceeding 200%. Compatible style pairs maintain high performance post-alignment, demonstrating that finetuning effectively mitigates style interference while enhancing controllability.

**Style Similarity Evaluation.** To quantify alignment with target styles, we compute the cosine similarity and mean squared error (MSE) between predicted and target style vectors. Table 2 shows that finetuned models generally outperform the multi-task baseline, achieving lower MSE and higher cosine similarity. This confirms that the policy adapts accurately to intended stylistic targets after a single trajectory-based adaptation.

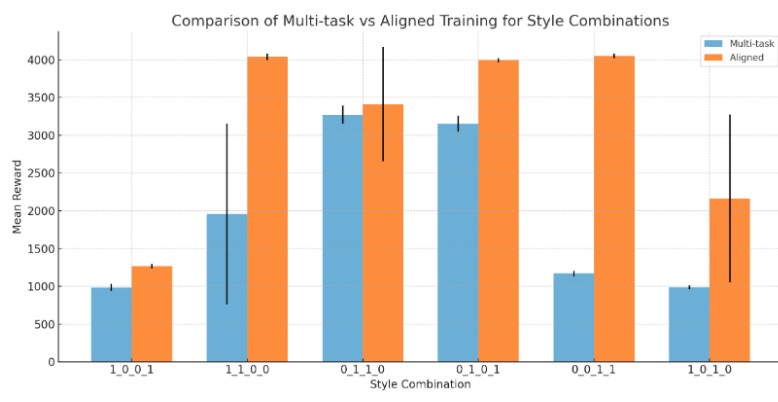

Figure 5: Average episode reward comparison across training methods for each style.

Table 2: Comparison of finetuning and multi-task learning on style expression quality (lower MSE and higher cosine similarity are better).

| Style Vec | Finetune-MSE | Finetune-Cos | Multi-MSE | Multi-Cos | MSE Win | Cos Win |
|-----------|--------------|--------------|-----------|-----------|---------|---------|
| 0_1_1_0 | 0.3266 | 0.6028 | 0.3690 | 0.5253 | ✓ | ✓ |
| 0_0_1_1 | 0.2244 | 0.9113 | 0.2723 | 0.7663 | ✓ | ✓ |
| 0_1_0_1 | 0.4228 | 0.3987 | 0.4034 | 0.4395 | x | x |
| 1_0_0_1 | 0.3797 | 0.4930 | 0.3455 | 0.5768 | x | ✓ |
| 1_0_1_0 | 0.3169 | 0.6563 | 0.3347 | 0.6159 | ✓ | ✓ |

**Generalization to Unseen Style Vectors.** We further assess zero-shot generalization using 200 randomly sampled style vectors from the continuous latent space (e.g., $[0.2, 0.3, 0.8, 0.3]$). Across these trials, our method achieves consistently lower MSE and higher cosine similarity compared to the multi-task baseline, indicating strong generalization beyond predefined styles. Representative trajectory rollouts are visualized in Figure 6, demonstrating that interpolated policies produce behaviors consistent with the desired mixtures.

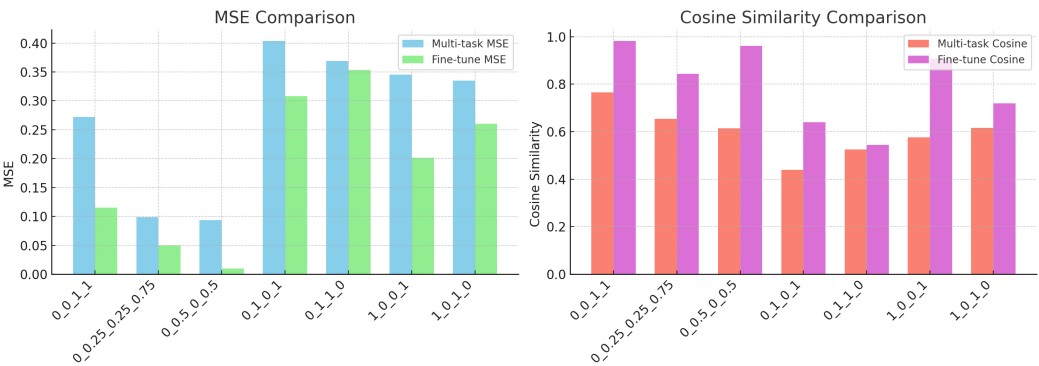

Figure 6: Cosine similarity and MSE for interpolated and unseen style combinations.

**Task Performance of Interpolated Styles.** Interpolated style vectors maintain or enhance task performance relative to single-style policies. For example, interpolations such as $[0.2, 0.3, 0.8, 0.3]$ and $[0.2, 0.4, 0.5, 0.6]$ achieve cumulative rewards of 3237 and 3222, surpassing the best single-style policy (Style 2: 2809). Figure 7 presents representative trajectories, illustrating that the model can integrate multiple stylistic influences without compromising task completion. These results suggest that interpolated styles can exploit synergies between base styles, offering a promising approach for balancing personalization and task effectiveness.

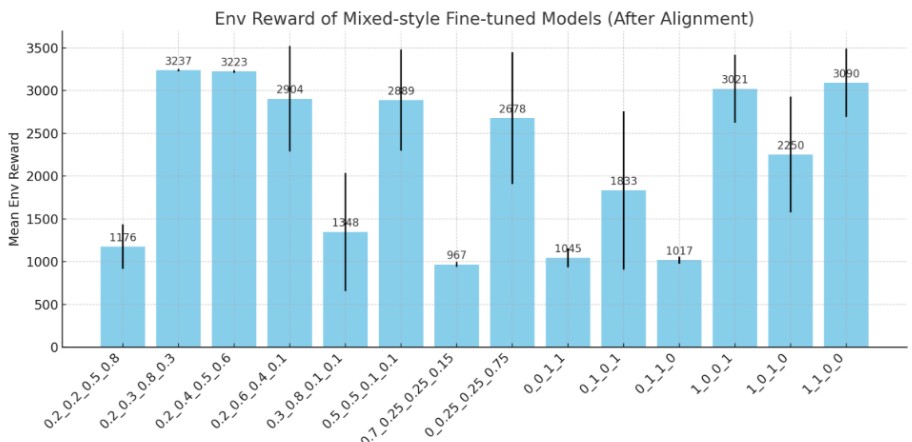

Figure 7: Task reward of finetuned interpolated style vectors.

## 4.5 Ablation Studies and Robustness

Compared to traditional methods, our approach enables more stable and efficient task completion, particularly in sparse-reward environments. For example, when evaluating the enjoyable-style agent under fixed style weights, baseline models exhibited high variance and occasional non-convergence. Moreover, directly adding style rewards often failed to elicit clear stylistic behaviors, even with carefully tuned weights. Detailed results are shown in Table 3.

Table 3: Performance Metrics Across Different Styles

| Style weight | Avg rew | Std | Avg predict pro | Accracy |
|---|---|---|---|---|
| 0.2 | 191.5 | 25.65 | [0.02, 0.02, 0.17, 0.73] | 0.15 |
| 0.6 | 169.5 | 16.12 | [0.02, 0.02, 0.22, 0.66] | 0.15 |
| 1.0 | 165.0 | 25.59 | [0.03, 0.02, 0.16, 0.76] | 0.13 |

We evaluated the effect of different learning rates on model performance (Table 4). A rate of 0.005 yielded the best trade-off between stability and reward. Overall, performance was relatively consistent across settings, indicating low sensitivity to the learning rate within a reasonable range.

Table 4: Performance with Different Learning Rates

| Value | Success Rate(%) | Avg rew | Convergence round |
|---|---|---|---|
| 1e-4 | 97.5 | 188.5 | 190 |
| 5e-5 | 98.0 | 189.5 | 170 |
| 1e-6 | 95.0 | 177.0 | 120 |

## 5 Conclusion, Limitations and Future Work

In this paper, we propose a framework for one-shot style personalization in reinforcement learning, enabling a pre-trained agent to adapt its behavior from a single demonstration. Our method achieves precise style alignment while maintaining task performance and generalizes to novel style mixtures through vector interpolation. Experiments on continuous control and cooperative multi-agent tasks show substantial improvements over baseline methods. While our study focuses on simulated control and game-like environments, future work could explore broader domains, including autonomous driving, human-robot collaboration, and language-based tasks, where fine-grained style alignment could enhance personalization and user-centric performance.

ETHICS STATEMENT

This work adheres to the ICLR Code of Ethics. In this study, no human subjects or animal experimentation was involved. All datasets used were sourced in compliance with relevant usage guidelines, ensuring no violation of privacy. We have taken care to avoid any biases or discriminatory outcomes in our research process. No personally identifiable information was used, and no experiments were conducted that could raise privacy or security concerns. We are committed to maintaining transparency and integrity throughout the research process.

REPRODUCIBILITY STATEMENT

We have made every effort to ensure that the results presented in this paper are fully reproducible. The experimental setup, including training steps, model configurations, and hardware details, is described in detail, and comprehensive pseudocode is provided to facilitate replication and verification. Additionally, the procedures for generating the self-created datasets are described in detail, ensuring consistent and reproducible evaluation results.

We believe these measures will enable other researchers to reproduce our work and further advance the field.

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

## A  LLM Usage

Large Language Models (LLMs) were used to aid in the writing and polishing of the manuscript. Specifically, we used an LLM to assist in refining the language, improving readability, and ensuring clarity in various sections of the paper. The model helped with tasks such as sentence rephrasing, grammar checking, and enhancing the overall flow of the text.

It is important to note that the LLM was not involved in the ideation, research methodology, or experimental design. All research concepts, ideas, and analyses were developed and conducted by the authors. The contributions of the LLM were solely focused on improving the linguistic quality of the paper, with no involvement in the scientific content.

The authors take full responsibility for the content of the manuscript, including any text generated or polished by the LLM. We have ensured that the LLM-generated text adheres to ethical guidelines and does not contribute to plagiarism or scientific misconduct.

## B  Data Collection Details

### Hopper Environment

In Hopper, each style is induced through modifications to the environment's reward function:

- **Smooth-Motion**: penalizing high-frequency actuation changes to promote smooth trajectories.
- **Energy-Saving**: penalizing large action magnitudes to minimize torque usage.
- **Posture-Stable**: reinforcing torso position and stability to encourage upright movement.
- **Speed-Oriented**: increasing rewards for forward velocity to favor rapid locomotion.

Each style generates 200 trajectories (total 800 trajectories, 400K timesteps), stored in HDF5 format.

### Overcooked Environment

In Overcooked, we design rule-based agents exhibiting different planning biases:

- **Selfish**: prioritizing personal reward, showing minimal path yielding or cooperation.
- **Cooperative**: emphasizing path negotiation, task handoffs, and minimizing partner conflict.
- **Enjoyable**: inconsistent or inefficient task planning, mimicking novice or entertainment-seeking users.
- **Efficient**: optimizing throughput regardless of coordination mode, often balancing solo and team play.

Each style yields 200 full episodes and 8000 sliced segments. Data is partitioned in a 4.8:1:1 ratio for training, validation, and testing.

## C  Styles of Human Preferences in Overcooked and Hopper

### C.1  Overcooked

We define four representative partner styles in the Overcooked environment, designed to capture diverse human preferences in social behavior and cooperation through qualitative annotations.

**Selfish**: Focuses on personal rewards, often ignoring partners. Tends to cause collisions and compete for high-reward tasks.

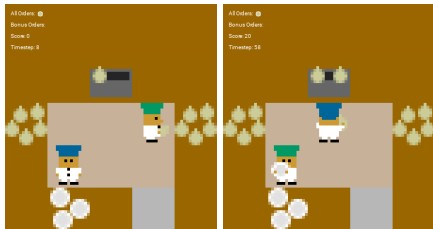

Figure 8: Selfish Behavior Examples

**Cooperative**: Prioritizes teamwork, avoids blocking, and helps complete others' tasks. Resolves path conflicts by yielding.

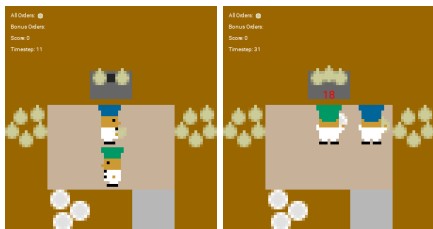

Figure 9: Cooperative Behavior Examples

**Enjoyable**: Often random or inefficient, typical of novices. Low success rate, unstable behavior, and frequent idle movement.

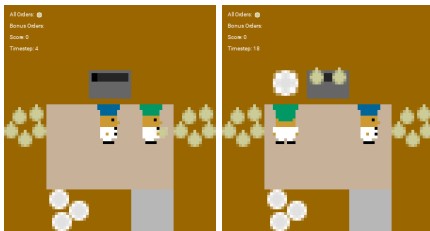

Figure 10: Enjoyable Behavior Examples

**Efficient**: Task-focused and productive. Balances actions without strong selfish or cooperative traits.

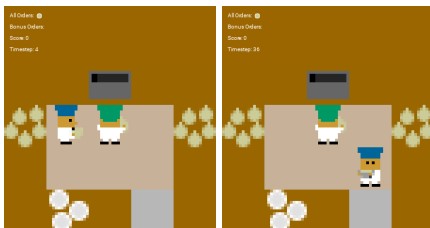

Figure 11: Efficient Behavior Examples

## C.2 HOPPER

We define four representative locomotion styles in the Hopper environment, designed to reflect diverse preferences in control strategy and gait characteristics through targeted reward shaping.

**Smooth-Motion**: Promotes fluid and natural hopping by penalizing high-frequency changes in joint torques. This results in consistent and graceful movement patterns with fewer abrupt shifts.

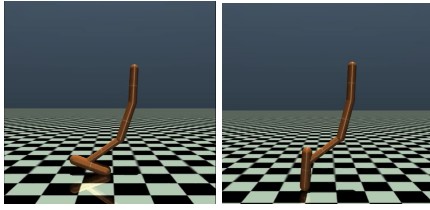

Figure 12: Smooth-Motion Hopper Behavior

**Energy-Saving**: Discourages excessive motor effort by penalizing large action magnitudes. The agent learns to move with minimal energy consumption, often resulting in smaller but more deliberate hops.

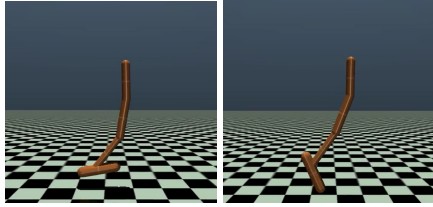

Figure 13: Energy-Saving Hopper Behavior

**Posture-Stable**: Reinforces upright torso posture by rewarding vertical alignment and penalizing excessive lean. The resulting gait emphasizes stability and body control, reducing the chance of falling.

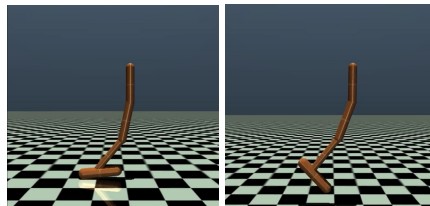

Figure 14: Posture-Stable Hopper Behavior

**Speed-Oriented**: Rewards high forward velocity, encouraging the agent to move quickly across the terrain. This style typically results in longer strides and more aggressive thrusts.

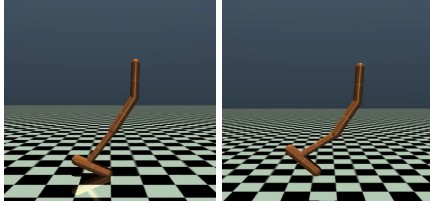

Figure 15: Speed-Oriented Hopper Behavior

# D EFFECTIVENESS OF PERSONALIZED ALIGNMENT FRAMEWORK

We evaluate the style alignment capability by conducting controlled fine-tuning experiments on four distinct styles. For each style, the target probability vector emphasizes the corresponding dimension (0.95 for target, 0.05 for others). Figure 16 shows the reward dynamics during training across 400 iterations, including composite reward and sparse environmental rewards. The consistent convergence patterns across styles indicate the robustness of our framework in balancing task performance and style alignment.

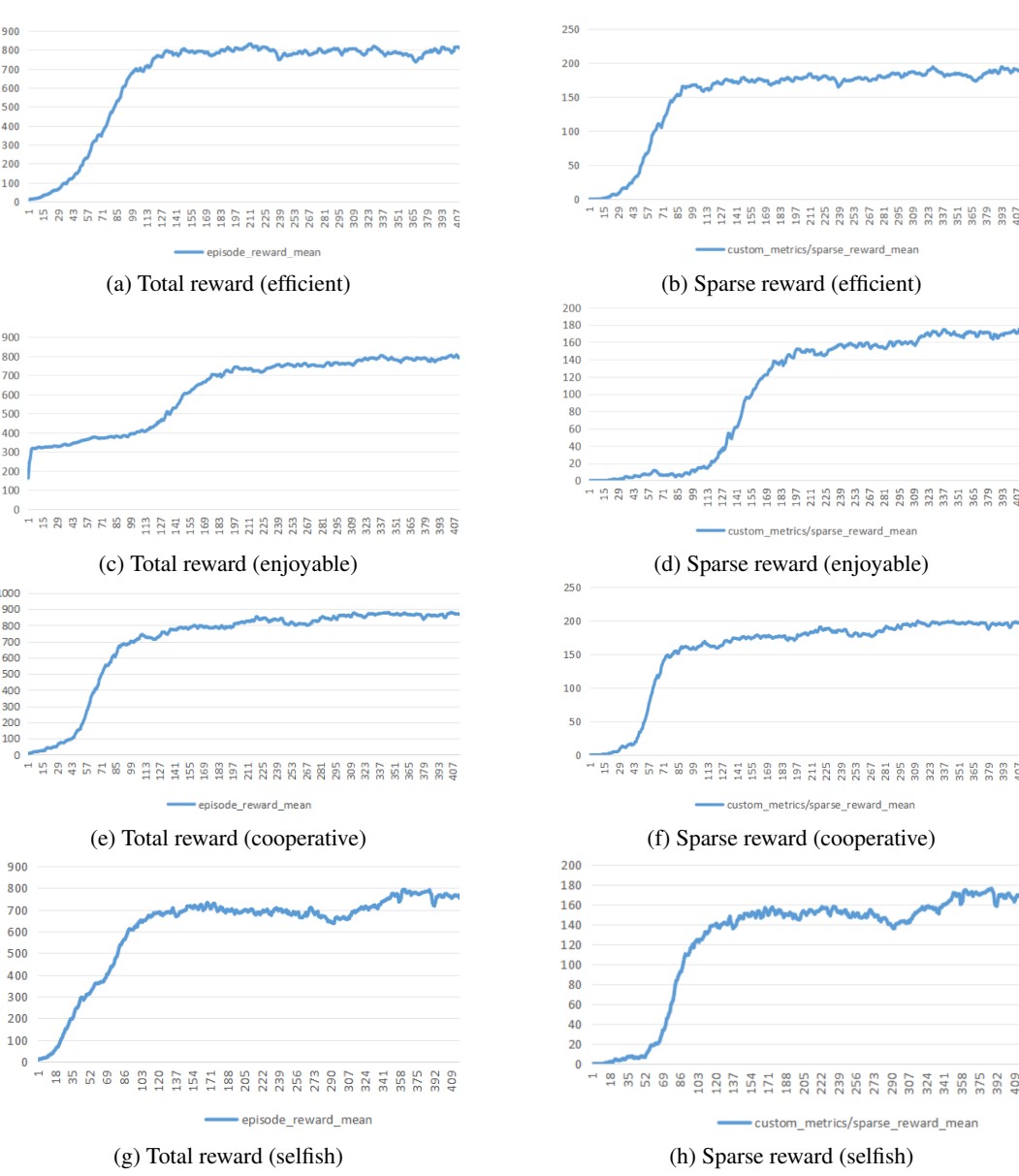

Figure 16: Training dynamics across four distinct styles through 400 iterations. (a,c,e,g) Evolution of composite rewards (environmental, shaping, and style components). (b,d,f,h) Progression of sparse environmental rewards. Consistent convergence patterns demonstrate framework robustness.

## E  IMPLEMENTATION DETAILS

All experiments were conducted on a Linux server with 4×RTX 2080 Ti GPUs (11GB VRAM) and 251GB RAM, using TensorFlow 2.10.0 with CUDA 10.1/cuDNN 7.6.5 acceleration. The implementation leveraged TensorFlow's native multi-GPU support, with datasets stored on a dedicated 3.6TB HDD partition to optimize I/O performance.

The trajectory window size was set to 40 steps, striking a balance between capturing sufficient temporal context for accurate style evaluation and maintaining timely responsiveness. The dynamic weighting factor for the style reward was linearly annealed from 0.1 to 1.0 over training steps ranging from 40 to 400. To enhance sensitivity to meaningful style differences while ensuring numerical stability, the raw cosine similarity scores were transformed via a squared scaling function with a coefficient of 0.4.

All experiments were conducted using multiple fixed random seeds $\{42, 56, 73\}$, with three independent runs performed for each configuration to ensure reproducibility.

## F  ADDITIONAL RESULTS

### F.1  T-SNE VISUALIZATION OF STYLE EMBEDDINGS

To complement the quantitative results in Table 1, we provide a qualitative analysis of the learned style representations. We project the output embeddings of the latent style discriminator into a 2D space using t-SNE.

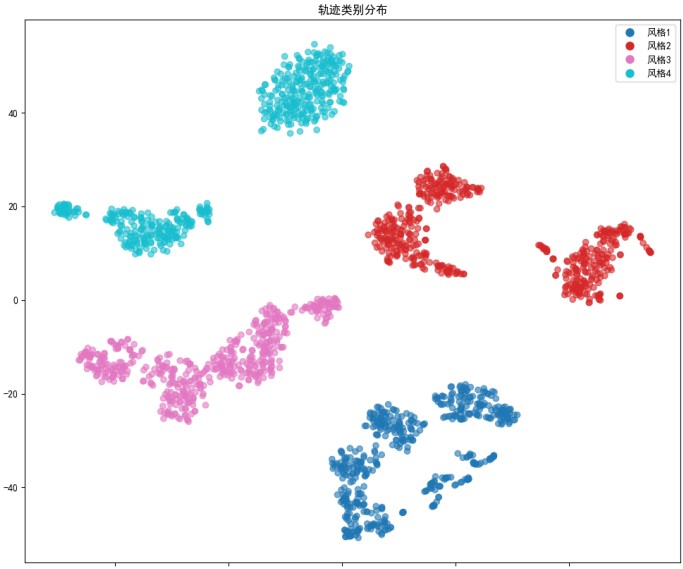

Figure 17: t-SNE visualization of style embeddings across trajectory classes. Each point represents a trajectory segment, colored by its ground-truth style label.

As shown in Figure 17, trajectories corresponding to different style classes form distinct and well-separated clusters, indicating that the discriminator learns highly discriminative embeddings. Meanwhile, within-class variability is preserved, reflecting the natural diversity of behaviors observed in real-world trajectories. These results complement the main text by providing an intuitive illustration of the latent space structure.

## F.2 SINGLE-STYLE BASELINE PERFORMANCE

Table 5: Average episode rewards (mean $\pm$ std) for single-style policies in the Hopper environment. These results provide reference performance for evaluating mixed-style policies.

| Style | Mean Reward $\pm$ Std |
|---|---|
| Style 0 | $1248.30 \pm 431.85$ |
| Style 1 | $3106.96 \pm 29.05$ |
| Style 2 | $2809.44 \pm 404.78$ |
| Style 3 | $1053.02 \pm 78.33$ |

# G  PSEUDOCODE

To facilitate reproducibility and clarity, we present the key components of our framework in pseudocode form. These include the training process of the Granular Ball-based style discriminator, the inference pipeline for trajectory style prediction, and the fine-tuning procedure for style-aligned reinforcement learning.

## G.1  TRAINING THE GRANULAR STYLE DISCRIMINATOR

Algorithm 1 outlines the procedure for training the Granular Ball-based style discriminator, which partitions trajectory features into interpretable style clusters ("balls") and builds a discriminator based on their geometric properties.

---
**Algorithm 1** TrainGranularModel

---
**Require:** Feature matrix $\mathcal{X}$, label vector $\mathcal{Y}$, model object $\mathcal{M}$
**Ensure:** Trained granular model $\mathcal{M}$
 1: Stack $\mathcal{X}$ and $\mathcal{Y}$ into dataset $\mathcal{D}$
 2: Generate initial granular balls $\mathcal{B} \leftarrow$ GENBALLSBALANCED($\mathcal{D}$)
 3: **for** each ball $b$ in $\mathcal{B}$ **do**
 4:     Compute center, radius, and majority label
 5:     Add to model
 6: **end for**
 7: Normalize radii using the median value
 8: **for** each pair of balls $(i, j)$ **do**
 9:     **if** balls $i$ and $j$ are overlapping **then**
10:         Mark larger ball for splitting
11:     **end if**
12: **end for**
13: **for** each marked ball **do**
14:     Apply KMeans to split into smaller sub-balls
15:     Replace original ball with sub-balls
16: **end for**
17: **for** each class label $y$ **do**
18:     **if** ball count for $y$ is too low **then**
19:         Generate additional granular balls for $y$
20:     **else if** ball count for $y$ is too high **then**
21:         Downsample balls to target count
22:     **end if**
23: **end for**
24: Train Random Forest on granular centers and labels
25: **return** trained model $\mathcal{M}$

---

## G.2 STYLE INFERENCE FROM A SINGLE TRAJECTORY

Algorithm 2 describes how to preprocess a raw trajectory, extract its features, and use the trained Granular Ball style discriminator to predict both its class and style probability distribution.

---

**Algorithm 2** Trajectory Style Prediction

---

**Require:** Trajectory $\mathbf{T}$, model $\mathcal{M}$, optional flag $v$
**Ensure:** Predicted class $c$ and probability vector $\mathbf{p}$
 1: Resize $\mathbf{T}$ to length 200 via padding or truncation
 2: Extract and scale features $\mathbf{f}_s \leftarrow \text{PREPROCESS}(\mathbf{T}, \mathcal{M})$
 3: $\mathbf{p} \leftarrow \mathcal{M}.\text{predict\_proba}(\mathbf{f}_s); \quad c \leftarrow \arg\max(\mathbf{p})$
 4: **if** $v$ is True **then**
 5:     Visualize trajectory and prediction
 6: **end if**
 7: **return** $c, \mathbf{p}$

---

## G.3 FINE-TUNING THE AGENT FOR PERSONALIZED STYLE

Finally, Algorithm 3 details how we fine-tune a pretrained SAC agent using our style reward mechanism, balancing environmental objectives with stylistic alignment to the target vector.

---

**Algorithm 3** Fine-tuning SAC Agent with Style Reward (Hopper)

---

**Require:** Pretrained SAC model $\mathcal{M}_0$, target style $\mathbf{s} \in \mathbb{R}^d$, discriminator $\mathcal{D}$
**Ensure:** Fine-tuned SAC model $\mathcal{M}_{\text{style}}$
 1: Wrap environment with STYLEHOPPERWRAPPER using $\mathbf{s}$ and $\mathcal{D}$
 2: **for** each timestep $t$ **do**
 3:     Sample action $a_t \sim \pi(s_t)$ and observe $s_{t+1}, r_t^{\text{env}}$
 4:     Store $(s_t, a_t, r_t^{\text{env}})$ in trajectory buffer
 5:     **if** trajectory buffer has $L$ steps **then**
 6:         Extract segment $\tau$ and compute $\hat{\mathbf{s}} \leftarrow \mathcal{D}(\tau)$
 7:         $r_t^{\text{style}} \leftarrow 1 - \frac{\|\hat{\mathbf{s}} - \mathbf{s}\|_2}{\sqrt{d}}$ {style alignment reward}
 8:         $r_t \leftarrow \lambda \cdot r_t^{\text{style}}$; increase $\lambda$
 9:     **else**
10:         $r_t \leftarrow r_t^{\text{env}}$
11:     **end if**
12:     Store transition $(s_t, a_t, r_t, s_{t+1})$ and update $\mathcal{M}_0$
13: **end for**
14: Save fine-tuned model $\mathcal{M}_{\text{style}}$
15: **return** $\mathcal{M}_{\text{style}}$

---

