# OpenReview forum: "One-Shot Style Personalization for RL Agents via Latent Discriminator"
_ICLR.cc/2026/Conference — Submitted to ICLR 2026_

### Official Review · Reviewer_7eLz · 2025-10-31

**Soundness:** 2
**Presentation:** 1
**Contribution:** 2
**Rating:** 2
**Confidence:** 4

**Summary:**

This paper proposes to utilize the GB method for dimensionality reduction and feature extraction on trajectory segments. The similarity of these extracted features is then employed as an auxiliary reward to fine-tune a PPO agent, with the goal of achieving style alignment.

**Strengths:**

1.  The paper is easy to follow, and the proposed method is straightforward and easy to understand.

**Weaknesses:**

1.  The proposed methodology exhibits low technical innovation and a lack of clear contribution. Leveraging GB-extracted features as an auxiliary reward to fine-tune PPO inherently results in very low flexibility. Specifically, introducing a new style necessitates the complete re-training of the policy.
2.  The definition of "style" is highly constrained. The paper does not clearly specify how style is defined. Based on the presented material, I must speculate that $N$ representative trajectories are manually selected, their latent features calculated, and these features define the representative styles. If this is the case, the four styles used in the experiments are entirely defined by only four trajectories, and the resulting policy diversity would be severely limited by the choice of these initial trajectories.
3. The experiments are too few and overly simple, consisting only of a single Mujoco locomotion task and one Overcooked game task.
4.  The experimental design appears unreasonable.
    * Table 1 aims to validate the capability of the style discriminator. However, this capability fundamentally belongs to the GB method itself and is not directly related to the authors' core contribution.
    * Figure 4 compares a Behavior Cloning (BC) policy, a standard RL policy, and the fine-tuned RL policy using an unspecified "reward" as the metric. I hypothesize this reward is the $R_{total}$ from Equation 3. If so, the poorer performance of the baselines is unsurprising and expected, as they were not trained to optimize this specific combined reward. (Furthermore, the fine-tuned RL policy does not appear to demonstrate a comprehensive advantage.)
5. Figure 17 appears to be a Chinese figure caption, English should be used.

**Questions:**

1.  Clarification on Style Definition and Latent Feature Acquisition:
    How do the authors specifically define "style"? For example, in concrete terms, how is the latent feature $z$ corresponding to "Speed-Oriented" style obtained? Does this involve manually selecting a single "Speed-Oriented" trajectory, or is it an aggregated feature derived from a collection of such trajectories? A clear explanation of the process for obtaining and representing the target latent features is essential for understanding the method's implementation and replicability.

---

### Official Review · Reviewer_cf5L · 2025-10-31

**Soundness:** 3
**Presentation:** 3
**Contribution:** 3
**Rating:** 4
**Confidence:** 3

**Summary:**

This paper introduces a one-shot style personalization framework for reinforcement-learning (RL) agents, enabling adaptation to human-preferred behavioral styles from a single demonstration without retraining. The method builds a latent style space using a Granular-Ball (GB)-based discriminator that clusters trajectories into interpretable style representations. A pretrained base policy is then conditioned by a style reward computed through cosine similarity between current and target style embeddings, dynamically balanced with environment rewards. Experiments on Hopper (MuJoCo) and Overcooked-AI demonstrate high style fidelity, task performance retention, and interpolation/generalization to unseen styles.

**Strengths:**

1. Novel framing of “one-shot style alignment.” The paper extends preference alignment to continuous stylistic adaptation—a timely and underexplored topic bridging RLHF and personalization.
2. Interpretability through Granular-Ball clustering. The GB discriminator provides semantic structure and robustness, addressing the usual opacity of learned latent spaces.
3. Strong empirical design. The experiments are diverse (single- and multi-agent), include interpolation and ablation studies, and report both style-fidelity and task-reward metrics, showing consistent improvements.

**Weaknesses:**

1. Limited comparison breadth. Baselines (only BC and Original RL) are weak. Missing comparisons to modern alignment methods weakens the empirical grounding.
2. How sensitive is the method to the number of granular balls or hyperparameters of the GB clustering? Does interpretability degrade with more styles?
3. The stability and transferability of the latent style space are not evaluated; it is unclear whether the learned embedding generalizes across tasks or environments.

**Questions:**

1. Can this approach adapt to natural language–specified styles (e.g., “drive more cautiously”) instead of trajectory examples?
2. How does the method behave when the style and task objectives conflict? Any convergence failures?
3. Could the latent style space support lifelong adaptation (continual addition of new user styles) without retraining the discriminator?
4. Safety and interpretability concerns are not discussed, especially regarding potential misuse or undesired style imitation.

---

### Official Review · Reviewer_Xr3U · 2025-11-03

**Soundness:** 3
**Presentation:** 3
**Contribution:** 3
**Rating:** 4
**Confidence:** 2

**Summary:**

This paper aims to achieve one-shot style alignment, which shapes agents to human-specified styles via a single example. Operationally, it  infers a latent style vector through a learned discriminator and adapts a pretrained base policy using a style reward signal during online interaction. Experiments are performed to demonstrate its efficacy on style alignment.

**Strengths:**

It formally and concisely defines the one-shot style personalization problem under investigation.

Algorithm tables and pseudocode are provided to delineate the workflows.

The writing is easy to follow with clear logics.

**Weaknesses:**

Authors noted `most existing alignment frameworks ... treat preferences as singular and static objectives`. Are there alignment methods that treat preference as fine-grained (i.e., non-single) \mathbf{or} dynamic objectives?  To my knowledge there would be some literature in terms of these topics, which should not be ignored in positioning this work's contribution. It would be necessary to discuss them and distinguish this work on this basis in introduction, which could precisely situate this work.

Authors noted `adapting an agent to a different preference typically requires substantial additional data and retraining`.  For LLM-based agent, what about in-context learning in this phenomenon? In-context learning should be effective to adapt a LLM to a different preference. Could this principle be extend to general agents?

Theoretical analysis on the proposed method seems to be lacking, which is critical to evaluate the technical quality of the proposed method.

The baselines to compare should be involved and introduced in detail.

The layout of Figure 3 should be improved in terms of its overall excessive size, the small font size on the axis, and the repetitive title in figure and caption.

The tables provided in the main paper should be improved in terms of information density, e.g., Table 1-4.

The reference style should be revised. Please differentiate between \cite and \citep.

There are some typos, e.g,  Li et al. Li et al. (2015), Christiano et al. Christiano et al. (2017).

**Questions:**

Please kindly see the weaknesses window.

---

### Official Review · Reviewer_jfuQ · 2025-11-03

**Soundness:** 2
**Presentation:** 1
**Contribution:** 2
**Rating:** 2
**Confidence:** 2

**Summary:**

This paper introduces a framework for one-shot style personalization in reinforcement learning, enabling the agent to adapt to a human-specified behavioral style from a single example. The approach learns an interpretable latent style space using a Granular Ball-based discriminator over trajectories, infers a target style vector from the example, and fine-tunes the policy online with a style reward that is dynamically weighted to preserve task performance. The method also supports smooth interpolation among styles, enabling generalization to unseen mixtures. Empirical studies on Hopper (MuJoCo) and Overcooked show precise style alignment, competitive or improved task rewards, and robustness relative to baselines.

**Strengths:**

This paper proposes a coherent method that turns style into a continuous, interpretable, and compositional space rather than a set of hard labels. By learning a style discriminator and using a simple, dynamically annealed style reward to fine-tune a competent base policy, the approach is practical and data-efficient (no large preference datasets) while giving users predictable control over mixtures of styles via interpolation. Empirically, it demonstrates strong style fidelity with maintained or improved task returns, stable adaptation compared to fixed-weight variants, and generalization to unseen mixtures across two distinct settings.

**Weaknesses:**

1. Core terms and procedures are underspecified or inconsistently defined. For example, M Avg / W Avg is used without definition; the pipeline does not state whether the discriminator’s probability vector is used directly for fine-tuning or if an argmax one-hot “base style” is taken first (Algorithm 2 returns both c and p, but downstream usage is not spelled out line-by-line); interpolation is described as summation then normalize, but the normalization convention (L1 vs L2) is not fixed across the paper.
2. Main baselines are Behavior Cloning and the unmodified original RL; multi-task is considered only in the interpolation analysis. There is no head-to-head against preference reward modeling or trajectory-level calibration baselines, which are central in related work and would better position this contribution.
3. The approach is trained/evaluated on engineered styles and pseudo-style labels (handcrafted behaviors), and relies on a supervised discriminator. It remains unclear how robust the pipeline is when styles are harder to separate or noisier, or when no labeled styles exist; the paper mentions GB-based pseudo-labeling “in the absence of annotated labels,” but does not provide corresponding ablations on such noisy/ambiguous scenarios.
4. Minor issue: Citations need to be fixed throughout the paper. In L465, 0.005 should be 5e-5.

**Questions:**

1. Is z_target a non-negative vector sums to 1? Is it always taken as the discriminator probability directly or do you ever apply argmax to produce a one-hot target before fine-tuning? Is L1 or L2 used in normalization?
2. How exactly were the continuous mixtures chosen for interpolation experiments? (If z_target is not necessarily nonnegative in Q1, then extrapolation is needed to be evaluated.)
3. In real-world application, styles are usually vague and noisy. What happens if two pre-defined styles are hard to tell apart (overlapping distributions) or if the number of styles K increases and labels become noisy?
4. Why is the proposed one-example adaptation important in the first place? If I understand correctly, the “one-example” is used during the training stage for fine-tuning rather than for rapid adaptation at inference time. With only a single demonstration, there is little justification to expect the model to achieve strong personalization; conducting prolonged training based on the information contained in a single data point is unreliable in empirical applications.

---

### Meta-Review · Area_Chair_3DjC · 2025-12-28

**Summary:**

This paper studies one-shot style personalization in reinforcement learning, aiming to ensure the agent adapting to a style using a single trajectory during online interaction with a learned discriminator. Several reviewers questioned about lack of clarity, technical details, and sufficient numerical tests. The paper also does not provide any theoretical results. The authors do not provide any response to the comments by the reviewer.

**Reviewer Concerns:**

The reviewers raise many concerns such as lack of clarity, technical details, sufficient numerical tests given the paper do not provide any theoretical contribution which I second. The authors do not provide any response.

**Reviewer Scores:**

The authors do not provide any response.

---

### Decision · Program_Chairs · 2026-01-26

Reject